# SRS2 is required for MUS81-dependent CO formation in *zmm* mutants

**Valentine Petiot** (ID)**, Floriane Chéron** (ID)¤**, Charles I. White** (ID)**, Olivier Da Ines** (ID)*

Institut Génétique Reproduction et Développement (iGReD), Université Clermont Auvergne, UMR 6293 CNRS, U1103 INSERM, Clermont-Ferrand, France

¤ Current address: INRAE, UMR 1095 INRAE – Université Clermont Auvergne Genetics, Diversity & Ecophysiology of Cereals, Clermont-Ferrand, France

* olivier.da_ines@uca.fr

## Abstract

Helicases are enzymes that use the energy derived from ATP hydrolysis to translocate along and unwind nucleic acids. Accordingly, helicases are instrumental in maintaining genomic integrity and ensuring genetic diversity. Srs2 is a multifunctional DNA helicase that dismantles Rad51 nucleofilaments and regulates DNA strand invasion to prevent excessive or inappropriate homologous recombination in yeast. Consistently, the deletion of Srs2 has significant consequences for the maintenance of genome integrity in mitotic cells. In contrast, its role in meiotic recombination remains less clear. We present here substantial evidence that SRS2 plays an important role in meiotic recombination in the model plant *Arabidopsis thaliana*. Arabidopsis *srs2* mutants exhibit moderate defects in DNA damage-induced RAD51 focus formation, but SRS2 is dispensable for DNA repair and RAD51-dependent recombination in somatic cells. Meiotic progression and fertility appear unaffected in *srs2* plants but, strikingly, the absence of SRS2 leads to increased genetic interference accompanied by increased numbers of Class I COs and a reduction in MUS81-dependent Class II COs. We propose that SRS2 plays a role in MUS81-mediated resolution of a subset of recombination intermediates into Class II CO. The absence of SRS2 would thus lead to the alternative channeling of these recombination intermediates into the Class I CO pathway, resulting in an increased proportion of Class I CO.

## Author summary

Helicases are enzymes that use ATP to unwind DNA. They play a crucial role in maintaining genomic stability. One such helicase, Srs2, is known to regulate homologous recombination in yeast by preventing excessive or inappropriate recombination events. While its function in mitotic cell division is well understood, its role in meiosis - the process that generates reproductive cells - remains less clear. We show here that SRS2 plays a significant role in meiotic recombination

**Data availability statement:** All relevant data are within the manuscript and its Supporting information files.

**Funding:** The author(s) received no specific funding for this work.

**Competing interests:** The authors have declared that no competing interests exist.

in the plant *Arabidopsis thaliana*. Although Arabidopsis plants lacking SRS2 do not exhibit major defects in DNA repair or fertility, they show modifications in meiotic crossover recombination patterns. Specifically, the absence of SRS2 shifts the balance between Class I and Class II crossovers, leading to an increase in Class I crossovers and a decrease in MUS81-dependent Class II crossovers. These findings provide new insights into the role of SRS2 in meiosis, suggesting that it has a role in the regulation of Class II crossover formation.

## Introduction

Homologous recombination (HR) is a high-fidelity mechanism of repair of DNA double-strand breaks (DSBs) that plays a vital role in preserving genomic stability and promoting genetic diversity [1–3]. In mitotically dividing cells, HR repairs DNA breaks caused by both environmental and endogenous factors and is crucial for the recovery of stalled or collapsed replication forks. In meiotic cells, HR is essential for proper chromosome segregation and creates genetic diversity among meiotic products [2–5].

The central feature of HR is the use of a homologous DNA molecule as a template to restore the original DNA sequence. This process begins with the formation of DNA DSBs followed by the resection of the 5'-ended strands of the DSBs, creating long 3' single-stranded DNA (ssDNA) overhangs. These ssDNA overhangs are then coated by the recombinase RAD51 in somatic cells, or by RAD51 and DMC1 in meiotic cells, forming a right-handed helical nucleofilament [6]. This nucleofilament initiates a homology search and promotes strand invasion of a homologous DNA template by the 3'-ended DNA strand(s), which are then extended through DNA synthesis. The resulting joint recombination intermediates can finally be resolved through different pathways to complete the process and restore chromosome integrity [1,2,7,8].

The DNA-strand invasion step catalyzed by the DMC1 and/or RAD51 nucleofilament is an important point of regulation for the fate of DSBs [6,9]. The stability of the intermediates thus formed and their subsequent resolution can strongly influence the formation of crossovers (COs) or non-crossovers (NCOs). The assembly/disassembly, stability and activity of the nucleoprotein filament are highly dynamic processes, tightly regulated by the coordinated actions of various positive and negative factors [6,9,10]. In particular, a number of ATP-dependent helicases are instrumental by disrupting various recombination intermediates. In yeast, one such helicase is Srs2 (**S**uppressor of **R**AD **S**ix-screen mutant **2**), which has multifunctional roles in DNA metabolism processes and plays a critical role in maintaining genome stability by resolving DNA structures that can lead to recombination or repair errors [11–14]. Playing both anti-recombination and pro-recombination roles, the absence of Srs2 induces pleiotropic recombination phenotypes [11–14].

The Srs2 helicase belongs to the superfamily 1 (Sf1) of helicases and shows structural and functional similarities to the bacterial UvrD helicase. The *Saccharomyces cerevisiae* Srs2 protein is well-characterized, exhibiting 3′ to 5′ helicase activity and is considered a prototypical anti-recombinase due to its ability to remove Rad51

from ssDNA, thereby preventing excessive or inappropriate HR events [15–17]. Consequently, *srs2* mutants exhibit a hyper-recombination phenotype as well as synthetic lethality with a number of other mutations affecting proteins involved in homologous recombination [12,13,15,16,18–24]. One current hypothesis proposes that Srs2 disrupt Rad51 filaments to channel recombination intermediates into non-crossover pathways (through Synthesis-Dependent Strand Annealing, SDSA), thus preventing CO in mitotic cells. Additionally, Srs2 has roles in other DNA metabolism processes, such as replication fork repair, post-replication repair via the template-switching pathway, non-homologous end joining [11–13] and even plays a role in DNA damage checkpoint-mediated cell cycle arrest [25,26].

Understanding of the role of Srs2 in meiosis is however much less complete. *S. cerevisiae* Srs2 is upregulated during meiosis [27] and its absence leads to delay in DSB repair and meiotic progression, reduced formation of both NCOs and COs, and a reduction in spore viability [18,27–31]. Overexpression of Srs2 also decreases CO and NCO formation, and reduces spore viability [27]. Interestingly, overexpression of Srs2 specifically disrupts the formation of Rad51 filaments, whereas Dmc1 filaments remain unaffected [27]. This is in accordance with the *in vitro* demonstration that Dmc1 directly inhibits the ATPase activity of Srs2, preventing it from translocating on ssDNA [14,32]. This inhibition could help promoting the formation of COs by Dmc1, which are essential for the faithful segregation of homologous chromosomes. Finally, Srs2 has recently been proposed to protect against the accumulation of aberrant recombination intermediates at the end of meiotic Prophase I [30,31]. Indeed, *srs2* mutants exhibit DNA damage accumulation at the end of Prophase I that are associated with the aggregation of Rad51 visible after the completion of meiotic recombination [30,31]. Thus, fine regulation of Srs2 is necessary for homologous recombination to proceed normally during meiosis in *S. cerevisiae*.

No meiotic phenotype of Srs2 is apparent in *Schizosaccharomyces pombe* [33–35]. However, this is likely the consequence of the presence of a second UvrD-type DNA helicase, called Fbh1 (lacking in *S. cerevisiae*). Like *S. cerevisiae* Srs2, Fbh1 is capable of removing Rad51 from DNA *in vitro*, and *fbh1* mutants exhibit Rad51 accumulation in meiotic cells and a strong reduction in spore viability [36,37]. Furthermore, *srs2 fbh1* double mutants are synthetic lethal supporting their redundant roles in recombination [38].

Surprisingly given its importance, studies on the role of Srs2 helicase in homologous recombination in multicellular eukaryotes are scarce. Srs2 appears to be conserved through evolution with homologs found in Viridiplantae, Heterokonts, and Metazoa [39,40]. Interestingly however, no SRS2 ortholog is found in fish and mammals, although functional homologs have been suggested [11–13,40].

A homolog of Srs2 has been identified in plants and in particular in the model plant *Arabidopsis thaliana* [39,40]. The *AtSRS2* gene is located on chromosome 4 (At4g25120) and encodes a protein of 1147 amino acids. The two functional domains of the protein (the ATP-binding domain and the C-terminal domain) are highly conserved [39,40]. *In vitro* characterization of Arabidopsis SRS2 has confirmed that it is a functional 3'-5' helicase capable of unwinding DNA [39]. Specifically, Arabidopsis SRS2 is capable of processing branched structures generated during SDSA, and also exhibits strand annealing activity, indicating a potential role during HR. However, although *Arabidopsis thaliana* SRS2 shows clear helicase activity *in vitro*, its function *in vivo* remains to be determined.

To date, the *SRS2* gene from the moss *Physcomitrella patens* is the only plant ortholog studied *in vivo* [41,42]. Moss *srs2* mutants do not exhibit major defects in homologous recombination, show no hypersensitivity to DNA damage induced by bleomycin, no effect on gene-targeting events and no fertility defects [41,42]. Thus, the role of the SRS2 helicase in homologous recombination in plants remains to be demonstrated.

We present here an analysis of SRS2 function in homologous recombination in the flowering plant, *Arabidopsis thaliana*. Our data show that SRS2 is dispensable for DNA repair and RAD51-dependent recombination in somatic cells, although *srs2* mutants exhibit moderate defects in RAD51 focus formation. No effects on fertility and meiotic progression are seen, but strikingly, the absence of SRS2 leads to both increased genetic interference and number of Class I COs. This is accompanied by a concomitant reduction of MUS81-dependent Class II COs. We thus propose that Arabidopsis SRS2 supports MUS81-mediated resolution of a subset of recombination intermediates into Class II COs.

## Results

### Isolation and molecular characterization of SRS2 T-DNA insertion mutants

As mentioned above, a homolog of the yeast Srs2 was found in the Arabidopsis genome [39]. Srs2 is a homolog of the bacterial UvrD helicase. It is a Sf1a helicase translocating in the 3'-5' direction along ssDNA. Sf1 helicases contain several archetypical helicase motifs and these are conserved in the *Arabidopsis thaliana* SRS2 homolog (Figs 1 and S1). We used AlphaFold structure prediction for Arabidopsis SRS2 (AF-D1KF50-F1-v4 model; Fig 1A) to compare its structure and domain arrangement with that of *Saccharomyces cerevisiae* Srs2 (ScSrs2; AF-P12954-F1-v4 model; Fig 1B) [43]. The AlphaFold model for Arabidopsis SRS2 gives a moderately confident prediction, with an average per-residue confidence score (pLDDT) of only 68.2. Specifically, the N- and C-terminal regions were very poorly predicted (pLLDT<50; orange color). In contrast, the core region exhibited high per-residue confidence score (pLDDT>90; blue color) for most residues (Fig 1A). The predicted aligned error plot also shows very accurate prediction of the relative position of most residues (Fig 1A), particularly within the conserved helicase motifs (colored in Fig 1B–E). Superimposition revealed strong structural similarity within these conserved helicase motifs (Fig 1D). Remarkably, all but one of the amino acids recently identified using AlphaFold as being in contact with DNA, and further shown to compromise the activity of budding yeast Srs2, are conserved in *Arabidopsis thaliana* SRS2 (Fig 1B–E; [44]). Notably, K41, F285, and H650, all instrumental for helicase activity of yeast Srs2 [44] are conserved in the Arabidopsis protein (K273, F518, and H856, respectively; Fig 1E). Altogether, these findings support the presence of conserved helicase activity of Arabidopsis SRS2. However, yeast Srs2 Y775 (V963 in Arabidopsis), which was recently shown to be essential for disrupting D-loops [44], is not preserved in Arabidopsis SRS2 (Fig 1B–E). A similar loss was observed in the *Physcomitrella patens* SRS2 homolog (the other plant SRS2 identified to date), where Y775 is also not conserved (S1B Fig). This suggests a conserved loss in the plant lineage and points to a potential functional loss of D-loop disruption activity in plant SRS2.

In order to study SRS2 function *in planta*, we searched for and characterized 3 T-DNA insertion mutant lines, thereafter named *srs2–1* (GABI_637C01), *srs2–2* (GABI_647B04), and *srs2–3* (SALK_039766; S2A Fig). The mutant alleles were verified by PCR and sequencing to determine the exact genomic position of the insertions (S2A and S2B Fig). In *srs2–1*, the T-DNA is inserted in exon 7. This insertion is associated with a deletion of 32 bp and is flanked by two T-DNA left borders in opposite orientations (S2A Fig). Sequence analysis of the T-DNA junction indicated that an in-frame stop codon is present immediately at the T-DNA left border (S2A Fig). In *srs2–2* and *srs2–3*, the T-DNA is inserted in introns 3 and 15, respectively. In *srs2–3*, the insertion is associated with a deletion of 29 bp and is flanked by two left borders (S2A and S2B Fig). Sequence analysis of the T-DNA junction indicated that an in-frame stop codon is present in intron 15 before the T-DNA integration (S2B Fig). Homozygous mutant lines were further analyzed by RT-PCR to confirm the absence of the respective transcript (S2C Fig). For all three mutants, no transcript was detected with primers surrounding the T-DNA insertion site, confirming the absence of full-length transcript and that no functional SRS2 protein can be produced (S2C Fig).

### SRS2 is dispensable for DNA repair and RAD51-dependent recombination in somatic cells

To assess the involvement of SRS2 in DNA repair, we tested the sensitivity of the *srs2–1, srs2–2 and srs2–3* mutants to the DNA-damaging agent Mitomycin C (MMC). MMC forms DNA interstrand crosslink adducts and subsequently DNA breaks that are repaired by RAD51-dependent homologous recombination. No hypersensitivity to MMC was observed in the mutants (Fig 2A and 2B). These findings are in striking contrast to those observed in yeast. Given that SRS2 is the only UvrD-type helicase found in the Arabidopsis genome and that it appears to be expressed in all tissues [45], we hypothesized that the absence of SRS2 is compensated by other helicases. Indeed, in budding yeast, *srs2* mutants show synthetic lethality when combined with mutation of other helicases such as Rad54, Sgs1/Recq4a, and Mus81 (reviewed in [12]). We thus generated the corresponding double mutants in Arabidopsis (*srs2 rad54, srs2 recq4a, srs2 mus81*, and

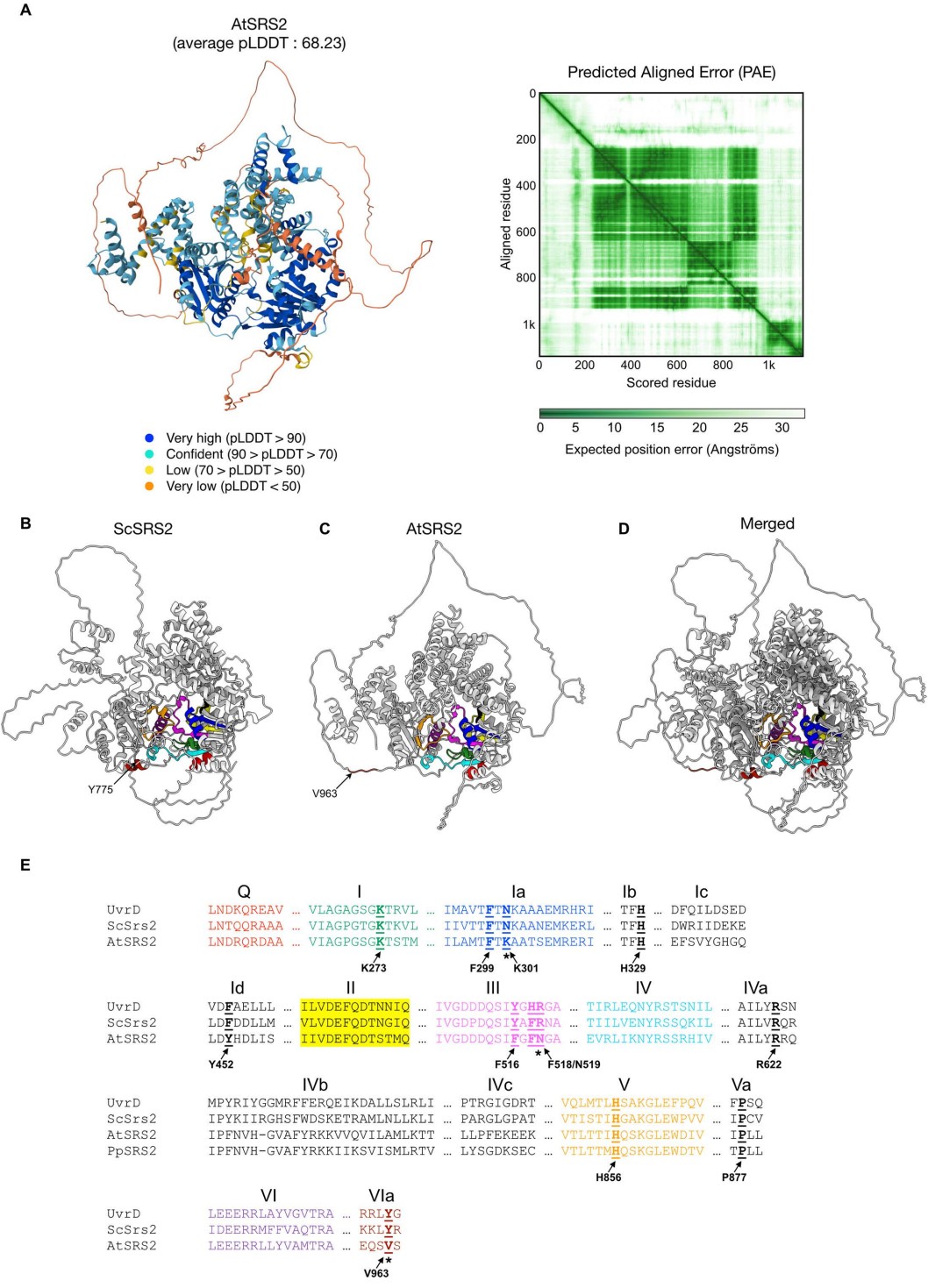

**Fig 1. AlphaFold-predicted structures of AtSRS2 and ScSRS2, and amino acid sequence comparison of UvrD, ScSRS2 and AtSRS2.** (A) AlphaFold-predicted structure of AtSRS2 with the associated Predicted Aligned Error (PAE). (B) AlphaFold-predicted structure of ScSrs2. Helicase domains are highlighted using the color scheme corresponding to the domains in Fig 1E. (C) AlphaFold-predicted structure of AtSRS2. Helicase domains are highlighted as in Fig 1E. (D) Superimposed predicted structures of ScSrs2 and AtSRS2 with helicase domains highlighted according to the color scheme in Fig 1E. (E) Amino acid residues shown to be essential for ScSrs2 activities are underlined in bold. Three of these amino acids are not conserved in Arabidopsis (neither identical nor similar) and are marked with an asterisk.

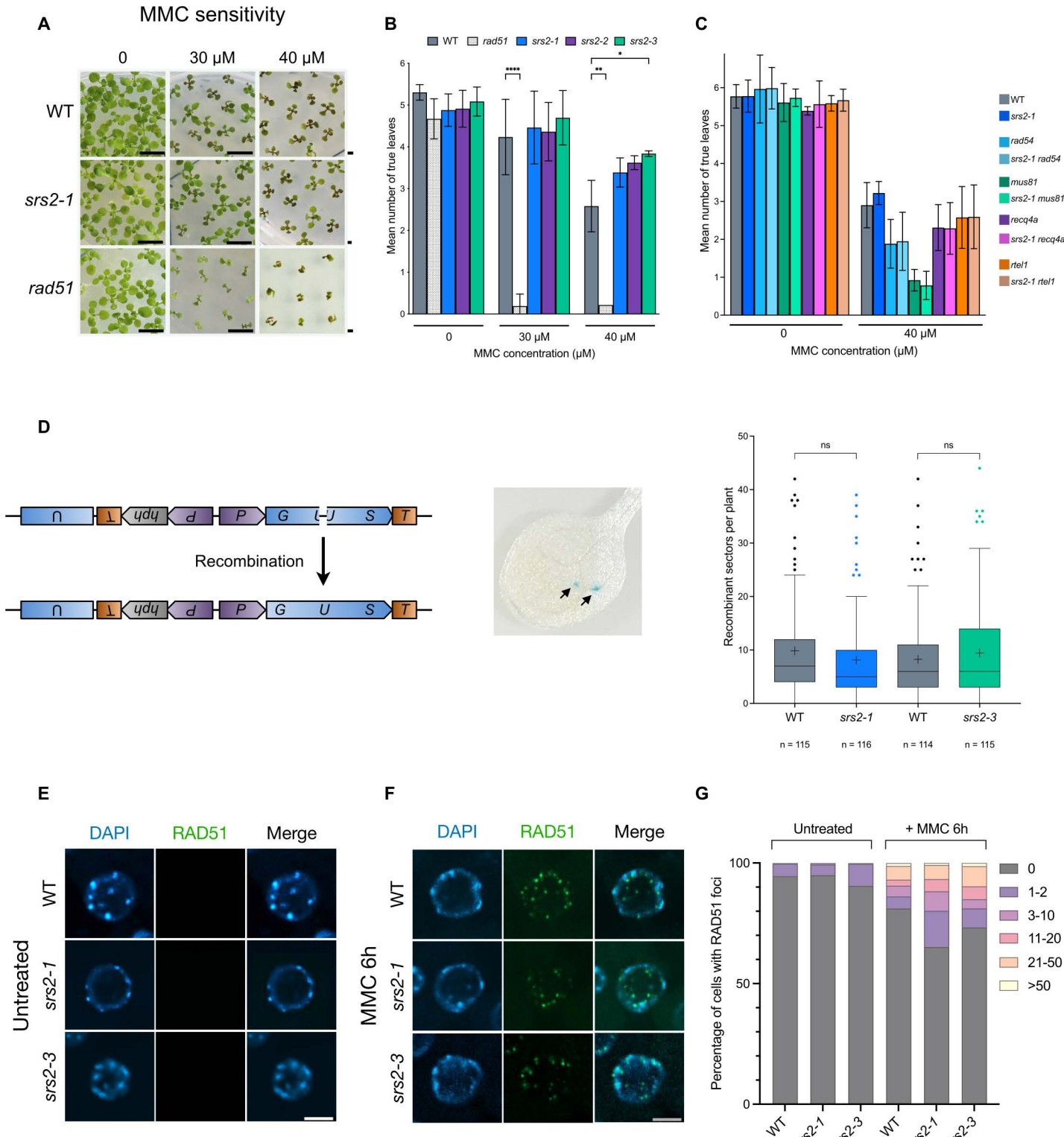

**Fig 2. AtSRS2 is dispensable for DNA repair and HR in somatic cells.** (A) Representative pictures of 2-week-old seedlings grown without (left), or with MMC (middle: 30 µM; right: 40 µM). (B) Mean number of true leaves per seedling. Data are shown as mean ± SD from 3 independent experiments, with 15-60 seedlings analyzed per genotype. Statistical analysis was performed using 2-way ANOVA test. * p-value < 0.05; ** p-value < 0.01; ****

p-value < 0.0001. *rad51* is RAD51-GFP transgenic line described in [72] (C) Mean number of true leaves per seedling of several helicase mutants and double mutant lines. Data are represented as mean ± SD of 3 independent experiments, with 15-60 seedlings analyzed per genotype. Statistical analysis was performed using 2-way ANOVA test. * p-value < 0.05. (D) Schematic representation of the IU.GUS reporter locus and an image showing blue spots (black arrows), indicating the assembly of functional GUS through HR events in an Arabidopsis leaf. The graph shows the quantification of spontaneous HR events (blue sectors) in somatic cells using the IU.GUS assay, with n indicating the number of seedlings analyzed. Each mutant was compared to wild-type (SRS2+/+, grey boxes) sister plants. Data are represented as Tukey box plots of 114-116 seedlings per genotype, with a "+" indicating the mean. Statistical analysis was performed using nonparametric Kruskal-Wallis test followed by Dunn's post hoc test for multiple comparisons. ns: non-significant; p-value > 0.05. (E-F) Immunolocalization of RAD51 in root tip nuclei of 5-days-old seedlings, either untreated (E) or treated with 30 µM MMC for 6h (F). Experiments were performed on *srs2-1* and *srs2-3* mutant lines. DNA is stained with DAPI (blue) and RAD51 foci (detected using an antibody against RAD51) are colored in green. Images are collapsed Z-stack projections of 3D image stacks. Scale bar: 5 µm. (G) Percentage of cells with 0, 1-2, 3-10, 11-20, 21-50, or > 50 RAD51 foci for each genotype, before and after MMC treatment.

*srs2 rtel1*) and tested their sensitivity to MMC. Neither the single nor the double mutants showed visible growth pheno-types without treatment. As expected, *rad54* and *mus81* single mutants displayed hypersensitivity to DNA damage (Fig 2C). However, no additional hypersensitivity was observed in the double mutants compared to the respective single heli-case mutants (Fig 2C). This contrasts strikingly with yeast double mutants [20], and suggests that SRS2 is dispensable for RAD51-dependent DNA repair in Arabidopsis.

To further investigate if SRS2 is involved in RAD51-dependent somatic homologous recombination (HR), we used the previously described IU.GUS recombination assay [46,47]. This system relies on the restoration of an interrupted, nonfunctional *beta-glucuronidase* (*GUS*) gene via HR, allowing quantification of somatic HR by histochemical staining. The reporter line was crossed with *srs2* mutants, and recombination was scored in the progeny homozygous for IU.GUS reporter. As anticipated, we did not observe any significant difference in HR event number per seedling between WT and *srs2* mutants (Fig 2D), further supporting the conclusion that SRS2 does not play a major role - or is functionally redun-dant with another helicase - in RAD51-dependent somatic HR.

A major role of Srs2 is dismantling Rad51 nucleofilaments through its ability to translocate along ssDNA. To check this, we performed RAD51 immunofluorescence on root apex nuclei of 5-day-old seedlings, treated or not with Mitomycin C (30 µM for 6 hours; Fig 2E–G).

As expected, very few RAD51 foci were detected without treatment, while numerous RAD51 foci were detected after MMC treatment in both wild-type and *srs2* mutants (Figs 2E–G and S3). Interestingly, the proportion of cells exhibiting RAD51 foci increases in *srs2* mutants (Fig 2G). In accordance with this, a moderate but significant increase in the mean number of RAD51 foci per nucleus was detected in *srs2* mutants (S3 Fig). In wild-type plants, we observed an average of 3.5 RAD51 foci per nucleus (± 10.8; n = 404), while *srs2–1* and *srs2–3* mutants had means of 4 (± 10.3; n = 345) and 4.1 (± 10.4; n = 391) foci per nucleus, respectively (S3 Fig). These results suggest that SRS2 may play a minor role in limiting RAD51 nucleofilaments and/or avoiding non-specific binding of RAD51 in somatic cells, without however having a major impact on DNA repair and RAD51-mediated recombination.

## Normal fertility and meiotic progression in the absence of SRS2

Homozygous *srs2* mutant plants develop normally and exhibit no fertility defects (Fig 3A). This is consistent with the sim-ilar pollen viability and apparent normal meiotic progression observed in *srs2* mutants, which show fully synapsed chro-mosomes at pachytene, 5 bivalents at metaphase I, normal chromosome segregation at anaphase I, and four balanced nuclei at the tetrad stage (Fig 3B and 3C). We estimated the chiasma number by counting the number of ring and rod bivalents at Metaphase I, assuming 2 or 3 crossovers, or 1 crossover for ring and rod bivalents, respectively. We counted an average of 9.2 chiasmata per cell in *srs2* mutants, not significantly different from the 8.8 chiasmata per cell in wild-type plants (Fig 3D). Finally, RAD51 and the synaptonemal complex transverse filament protein ZYP1 appear to load normally in *srs2* mutants (Fig 3E and 3F). Thus, the absence of SRS2 does not detectably affect meiotic progression, nor total chiasma numbers.

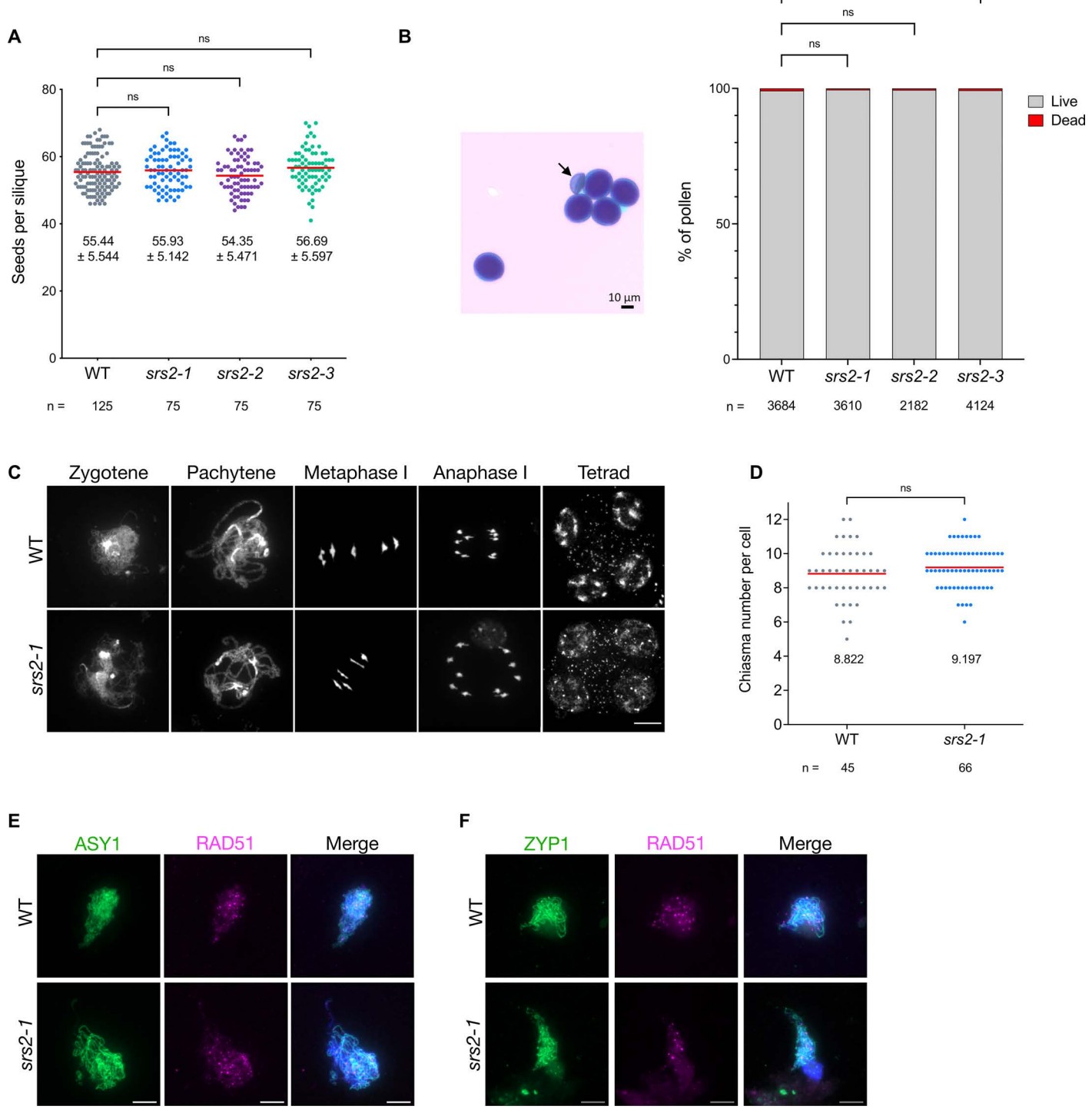

**Fig 3. AtSRS2 is dispensable for normal fertility and meiotic progression.** (A) Plant fertility was measured by counting the number of seeds per silique, with n indicating the number of siliques analyzed. Data are represented as mean (red line) ± SD, with each dot representing one silique. Statistical analysis was performed using Kruskal-Wallis test followed by Dunn's post hoc test for multiple comparisons. ns: non-significant. (B) Representative image of pollen viability assessed using Alexander staining in WT and *srs2* mutant lines. Viable pollen grains are stained purple, while dead pollen (indicated by a black arrow) appear empty and green. Data are represented as stacked bars showing the percentage of viable (grey) and

non-viable (red) pollen grains. n indicates the number of pollen grains counted for each genotype. Statistical analysis was performed using 2-way ANOVA test. ns: non-significant. (C) DAPI-stained male meiotic nuclei showing wild-type like meiotic progression in *srs2-1* mutant. Scale bar: 10 µm. (D) Chiasma number per cell at metaphase I stage, with n indicating the number of cells analyzed. Data are represented as mean (red line), with each dot representing one meiocyte. Statistical analysis was performed using unpaired t-test. ns: non-significant; p-value > 0.05. (E) Co-immunolocalization of RAD51 (magenta) and the chromosome axis protein ASY1 (green) on leptotene/zygotene meiotic chromosome spreads. Scale bar: 5 µm. (F) Co-immunolocalization of RAD51 (magenta) and ZYP1 (green) on zygotene/pachytene meiotic chromosome spreads. Scale bar: 5 µm.

In *S. cerevisiae*, Dmc1 is a potent inhibitor of Srs2 activity during meiosis [32]. Additionally, Srs2 specifically dismantles Rad51 nucleofilament, while recombination intermediates containing Dmc1 are not affected [27]. To test the hypothesis that Arabidopsis SRS2 specifically acts on RAD51-dependent recombination intermediates, we analyzed meiotic progression in the absence of both DMC1 (RAD51 becomes active) and SRS2. Meiotic progression of the *dmc1 srs2–1* double mutant closely resembles that of the *dmc1* mutant, in which RAD51 repairs meiotic DSB without forming interhomolog CO (S4 Fig). This suggests that SRS2 has no major effect on RAD51-mediated meiotic DSB repair in Arabidopsis meiosis, even in the absence of DMC1 (S4 Fig).

### Increased genetic interference in the absence of SRS2

We next sought to analyze more closely the impact of SRS2 on meiotic recombination by measuring meiotic CO rates in genetic intervals defined by markers conferring fluorescence in pollen grains (Fluorescent-Tagged-Lines, FTLs, [48]). When combined with mutation of the *QUARTET1* gene (*qrt*), which prevents separation of the four pollen grains, these FTL lines allow direct measurement of recombination between the linked fluorescent markers, and estimation of genetic interference by scoring pollen fluorescence in tetrads. We determined CO rates in two adjacent FTL intervals: I1bc (comprising I1b and I1c) on the left arm of chromosome 1, and I2fg (comprising I2f and I2g) on the long arm of chromosome 2, in both wild-type and *srs2* mutant plants (Figs 4B–E, S5 and S1 Table). No difference in recombination frequency was observed for either interval in *srs2–1* mutants (Fig 4A and 4B). Remarkably, however, genetic interference was substantially increased in the *srs2–1* mutant in both the I1bc and I2fg intervals (although this increase was not statistically significant for I2fg, likely due to the very short size of the interval, which limits the detection of a sufficiently large number of double CO events; Fig 4C–E). Specifically, interference increased from 0.53 in wild-type to 0.65 in *srs2–1* for I1bc, and from 0.8 in wild-type to 0.93 in *srs2–1* for I2fg (Fig 4C–E). Accordingly, the number of double COs in these intervals was significantly lower in the *srs2–1* mutants (Fig 4D and 4E). These findings were confirmed in *srs2–3* mutant for the I1bc interval, which also showed no change on CO frequency but an increased genetic interference (increasing from 0.53 in wild-type to 0.63 in *srs2–3* mutants; S5 Fig). Thus, genetic interference appears to be increased in the absence of SRS2. We note that this increase may not reflect a direct effect of SRS2 on genetic interference. Rather, it could result from a shift in the ratio of Class I to Class II CO, as previously observed in the *mus81* mutant, which shows increased genetic interference associated with reduced Class II CO formation [49].

### Altered balance of Class I and Class II CO in *srs2* mutants

Class I CO are sensitive to interference, while Class II CO are not [50]. Since the total number of CO is unchanged in *srs2* mutants compared to wild-type, the observed increase in genetic interference could result from a higher proportion of Class I COs, which are subject to interference. To test this hypothesis, we performed co-immunostaining of MLH1 and HEI10 on diakinesis-staged meiocytes. We counted an average of 9.2 MLH1-HEI10 co-foci in s*rs2–1* mutants (n = 35), significantly more than the 8.4 MLH1-HEI10 co-foci in wild-type meiocytes (n = 30; Fig 4F and 4G). This result was confirmed by MLH1 immunofluorescence on diakinesis-staged meiocytes (S6A Fig and S2 Table). A significant increase of MLH1 foci is seen in both *srs2–1* (12.2 foci, n = 55) and *srs2–3* (12.3 foci, n = 23) mutants compared to the wild-type (11.4 foci, n = 50; S6B Fig). The absence of SRS2 thus leads to an increase in Class I CO numbers.

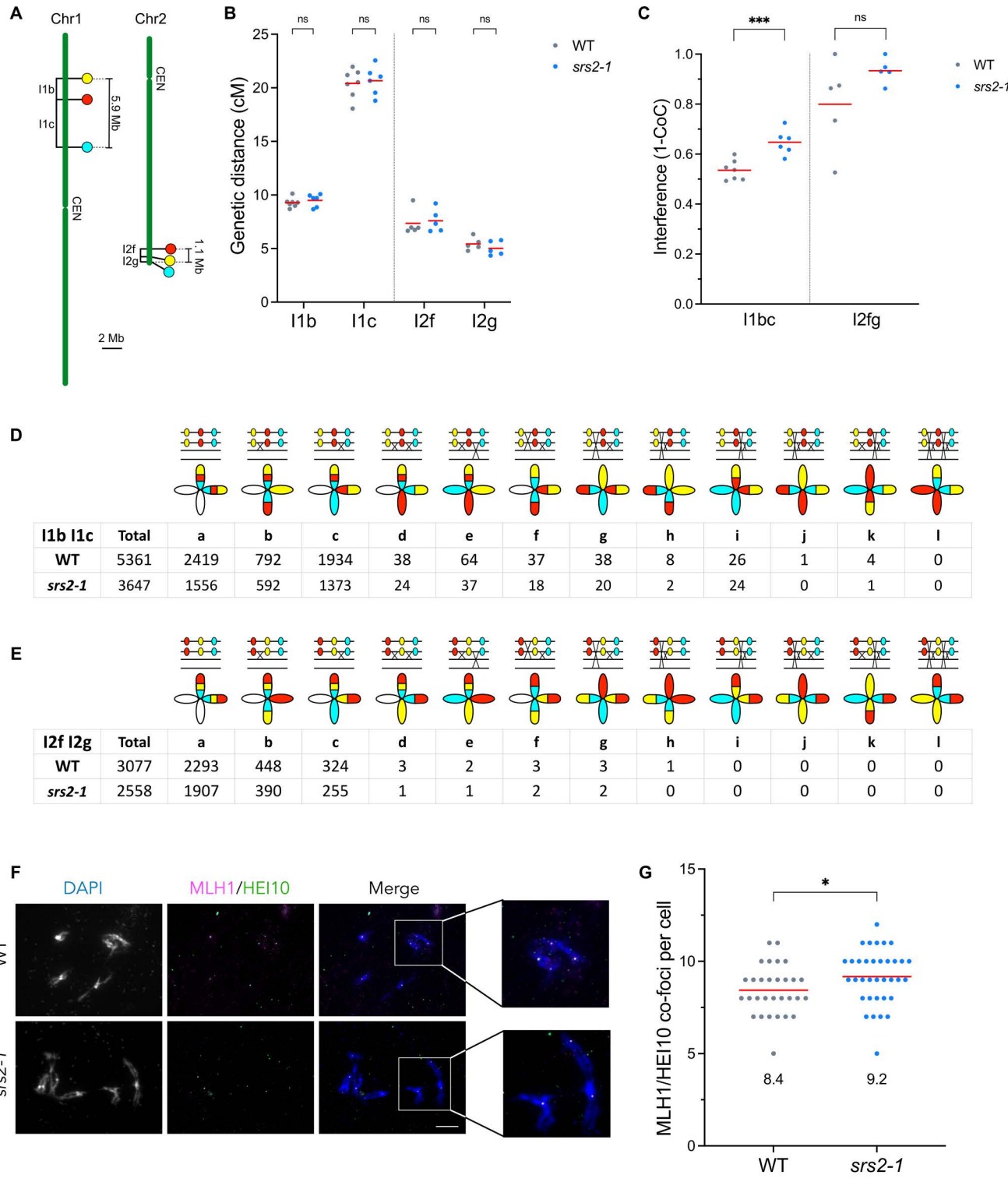

**Fig 4. Increased genetic interference and Class I CO in *srs2* mutants.** (A) Schematic representation of the localization of fluorescent markers for I1bc and I2fg lines. Physical distances of the intervals are indicated. CEN: centromere. (B) FTL crossover frequency in I1bc and I2fg intervals in wild-type (grey) and *srs2-1* (blue). CO frequency within I1bc and I2fg intervals is presented as the genetic distance. Each mutant is compared to wild-type

(WT) sister plants (SRS2 + / +, in grey). Data are represented as mean (red line), with each dot representing one plant. Statistical analysis was performed using Z-test. ns: non-significant; p-value > 0.05. (C) Crossover Interference within I1bc and I2fg intervals in WT and *srs2-1* mutants. Each mutant is compared to wild-type sister plants that are SRS2 + / + (in grey). Data are represented as mean (red line), with each dot representing one plant. Statistical analysis was performed using Z-test. P-value for I2fg interference: 0.1461. ns: non-significant; p-value > 0.05; *** p-value < 0.001. (D-E) Number of tetrads observed in wild-type and *srs2-1* for (D) I1bc and (E) I2fg. A schematic representation of the corresponding CO events is shown above each class of tetrad. (F) Representative images of MLH1/HEI10 co-immunolocalization on diakinesis-staged male meiocytes in WT and *srs2-1* mutant. Scale bar: 10 μm. (G) Number of MLH1/HEI10 co-foci per cell, with n indicating the number of cells analyzed. Data are represented as mean (red line), with each dot representing one individual cell. Statistical analysis was performed using Mann-Whitney test. * p-value < 0.05.

This being so, the absence of an overall increase in the number of CO would imply a corresponding reduction in the number of Class II CO in the absence of SRS2.

It is possible to quantify numbers of Class II CO in *zmm* mutants, which lack Class I COs. We thus crossed *srs2–1* mutant plants with the *zmm* mutants *zip4* and *msh5* and counted the number of bivalents at Metaphase I. As expected, we observed an average of 1.5 bivalents per cell in *zip4* and *msh5* mutant meioses (n = 49 and 37, respectively; Fig 5A–C). Remarkably, both the *zip4 srs2* and *msh5 srs2* double mutants exhibited significantly reduced numbers of bivalents per cell with 1.05 (n = 42) and 1.04 (n = 68) bivalents respectively per meiosis (Fig 5A–C). Numbers of Class II CO are thus reduced in the absence of SRS2.

## The absence of SRS2 affects MUS81-dependent Class II COs

The formation of Class II COs involves the action of structure-specific endonucleases, notably MUS81. To further confirm this impact of the absence of SRS2 on Class II COs, we tested its dependence on MUS81. We thus sought to assess whether the absence of SRS2 could affect MUS81-dependent Class II COs. *zip4 srs2–1* plants were crossed with *mus81* plants to produce *zip4*, *zip4 srs2–1*, *zip4 mus81*, and *zip4 srs2–1 mus81* mutants, and numbers of bivalents were counted at Metaphase I in these plants (Fig 5D–F). As expected, *zip4 mus81* have significantly fewer bivalents than the *zip4* single mutant (0.84 bivalent/cell, n = 77; versus 1.34 bivalent/cell, n = 56; [49,51,52]). We also confirmed that *zip4 srs2* exhibits fewer bivalents per cell (0.92, n = 56). Strikingly, no difference in the numbers of bivalents was seen between *zip4 srs2 mus81* triple mutant, *zip4 mus81,* and *zip4 srs2* double mutants (Fig 5D–F). SRS2 and MUS81 are thus epistatic for this phenotype, in agreement with a role for SRS2 in the Class II CO pathway.

## Discussion

Plants possess a SRS2 homolog, and *in vitro* studies of Arabidopsis SRS2 show that it is a functional 3'-5' helicase capable of unwinding DNA [39]. In accordance with this, key amino acids essential for Srs2 functions (e.g., Rad51 stripping) in budding yeast are conserved in Arabidopsis and moss SRS2 [44]. However, we note that Srs2 Y775 (V963 in Arabidopsis), essential for disrupting D-loops *in vitro* [44], is not conserved in Arabidopsis and Physcomitrella (Figs 1 and S2). This suggests a potential functional divergence or loss among plant SRS2 homologs, and raises questions about the *in vivo* role of plant SRS2. To clarify this, we isolated Arabidopsis mutants lacking SRS2 and analyzed its role in mitotic and meiotic recombination.

## SRS2 is dispensable for RAD51-dependent recombination in somatic cells

Our data show that the absence of Arabidopsis SRS2 has no detectable effect on sensitivity to DNA damaging agent Mitomycin C and RAD51-dependent recombination (Fig 2A,2B and 2D). This is in agreement with data in moss *Physcomitrella patens* where *srs2* mutants do not exhibit major defects in somatic homologous recombination [41,42] but in striking contrast with data in yeast where *srs2* mutants are highly sensitive to DNA-damaging agents and display a hyper-recombination phenotype [12,13,15,16,18–24]. Moreover, yeast *srs2* mutants are synthetic lethal when combined with other mutations that affect DNA repair and recombination [12,13,19,20,22]. Synthetic lethality of such *srs2* double mutants is suppressed by

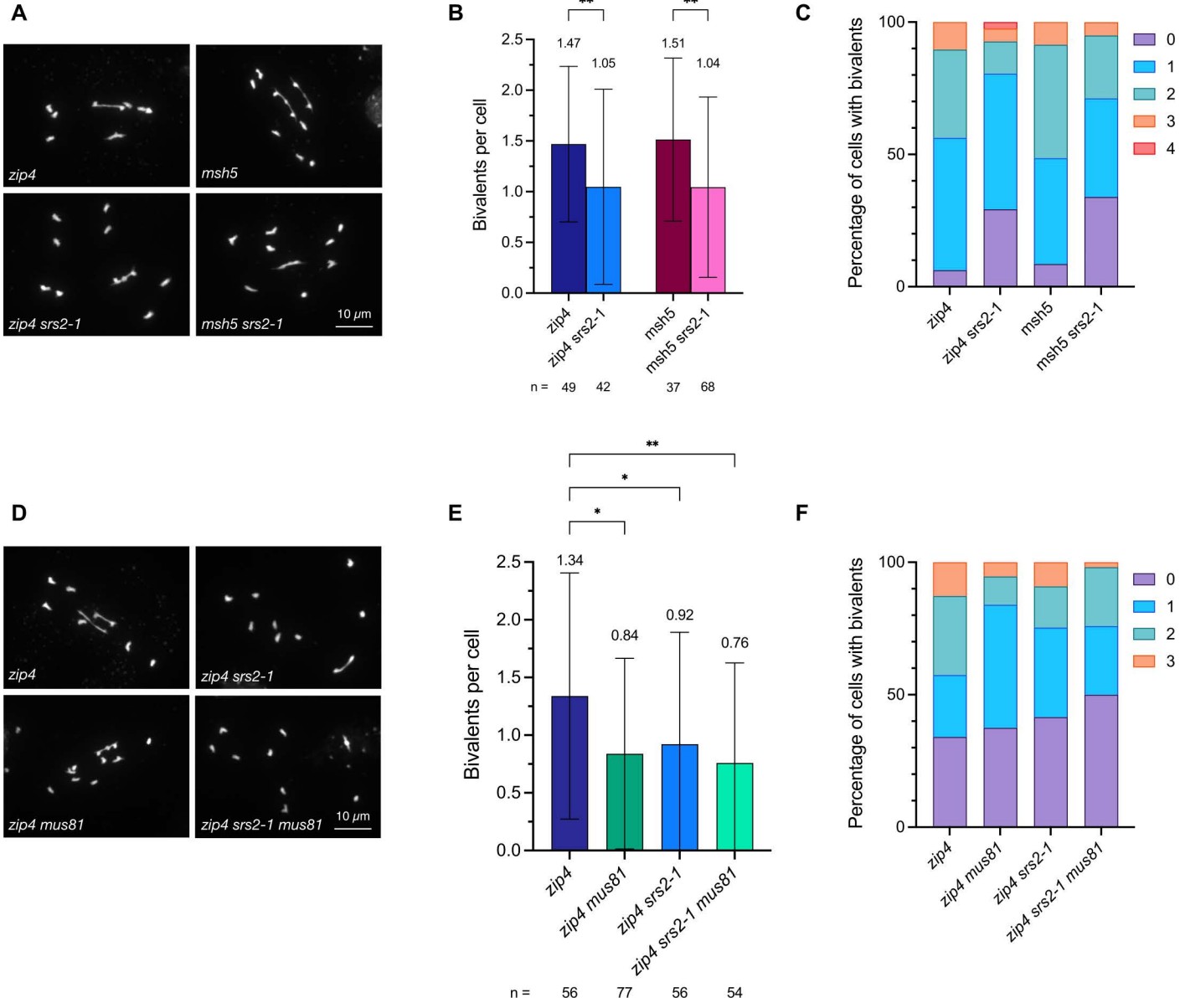

**Fig 5. Reduced MUS81-dependent Class II COs in *srs2* mutants.** (A) Representative images of metaphase I-staged male meiocytes in *zip4*, *zip4 srs2-1*, *msh5*, and *msh5 srs2-1* mutants. Scale bar: 10 μm. (B) Number of bivalents per cell in *zip4*, *zip4 srs2-1*, *msh5*, and *msh5 srs2-1* mutants. Data are represented as mean±SD, with n indicating the number of cells analyzed. Statistical analysis was performed using Mann-Whitney test. ** p-value<0.01. (C) Percentage of cells with 0, 1, 2, 3, or 4 bivalents in *zip4*, *zip4 srs2-1*, *msh5*, and *msh5 srs2-1* mutants. Number of cells analyzed is the same as in (B). (D) Representative pictures of metaphase I-staged male meiocytes in *zip4*, *zip4* mus81, *zip4 srs2-1*, and *zip4 srs2-1 mus81* mutants. Scale bar: 10 μm. (E) Number of bivalents per cell meiocytes in *zip4*, *zip4* mus81, zip4 *srs2-1*, and *zip4 srs2-1* mus81 mutants. Data are represented as mean±SD, with n indicating the number of cells analyzed Statistical analysis was performed using Kruskal-Wallis test followed by Dunn's post hoc test for multiple comparisons. * p-value<0.05; ** p-value<0.01. (F) Percentage of cells with 0, 1, 2, or 3 bivalents in *zip4*, *zip4* mus81, zip4 *srs2-1*, and *zip4 srs2-1 mus81* mutants. Number of cells analyzed is the same as in (E).

eliminating homologous recombination, suggesting that accumulation of toxic recombination intermediates is the cause of the lethality [11,13,44,12,53,54]. We thus tested synthetic genetic interaction between Arabidopsis SRS2 and several other helicases identified in yeast, such as RAD54, MUS81, or RECQ4A (Sgs1), as well as RTEL1 (putative functional homolog of SRS2 in mammals, also present in plants). However, none of the combinations tested led to an obvious phenotype, either under standard growth conditions or in response to induced DNA damage (Fig 2C). This suggests that the absence of Arabidopsis SRS2 does not lead to the accumulation of toxic recombination intermediates in the background tested, or alternatively, that this is compensated for by another helicase (or mechanism) that has not been tested in this study.

Interestingly however, we observed a slight increase in DNA damage-induced RAD51 foci in absence of SRS2 (S3 Fig). This moderate increase suggests that SRS2 could play a minor role in dismantling RAD51 nucleofilaments in somatic cells following DNA damage induction. This is in accordance with data in yeast where Srs2 is well-known for its anti-recombination function. It disrupts the RAD51 nucleofilament to promote the SDSA pathway and limit the number of CO in somatic cells, and also to prevent toxic or untimely HR events [15–17,27,55–57]. A recent study using a functionally tagged RAD51 to track RAD51 nucleofilament formation in living cells showed a moderate effect of *srs2* deletion on RAD51 nucleofilament formation [58]. Indeed, more functionally tagged RAD51 structures could be detected in a *S. cerevisiae srs2* mutant 2 hours after induction of a unique irreparable DSB in haploid cells. Interestingly, this increase was associated with brighter and longer RAD51 filaments in *srs2* mutants and (nearly) disappeared 4 hours after induction of the DSB [58]. Thus, in agreement with the known role of Srs2 in yeast, we hypothesized that Arabidopsis SRS2 could regulate RAD51 nucleofilaments. In the absence of SRS2, the nucleofilament is still formed, but more RAD51 could assemble along the filament leading to brighter, more readily detectable RAD51 foci (Fig 2E and 2F). Although this does not appear to affect repair of MMC-induced DNA damage or spontaneous RAD51-dependent HR events, it could impact the channeling of recombination events into different repair pathways. Alternatively, the absence of SRS2 could lead to non-specific accumulation of RAD51 at undamaged sites. However, we do not favor the latter hypothesis since we did not detect RAD51 foci in *srs2* mutants in absence of DNA damage.

## The absence of SRS2 affects Class I/Class II CO balance in meiotic recombination

Studies in budding and fission yeast have shown that Srs2 is highly expressed in meiosis and that absence of Srs2 reduces spore viability and decreases both COs and NCOs in meiosis [18,27–31,20]. Moreover, Srs2 protects against the accumulation of aberrant recombination intermediates at the end of meiotic Prophase I, as seen in the aggregation of Rad51 at late Prophase I in *srs2* mutants [30,31]. Interestingly, Srs2 appears to specifically affect Rad51-bound recombination intermediates since (i) Dmc1-bound intermediates are not affected, and (ii) Dmc1 inhibits Srs2's ATP hydrolysis activity [29,32]. These data suggest important functions of Srs2 in meiotic recombination.

In multicellular eukaryotes, evidence for a role of SRS2 homologs in meiosis remained to be demonstrated. Mammals have no ortholog of SRS2, although a number of functional homologs are known (RECQ5, PARI, RTEL1, BLM, FBH1) [2,12,40]. Our study highlights a role for SRS2 in meiotic recombination in Arabidopsis, and suggests that SRS2 is required for the formation of a subset of Class II COs, with evidence pointing towards an epistatic role of SRS2 with MUS81. Arabidopsis SRS2 is more strongly expressed in meiotic tissues than in somatic tissues [45,59] and we show here that the absence of SRS2 leads to a shift in the ratio of Class I to Class II COs (more Class I COs and less Class II COs), without affecting total chiasma number (Figs 4G, 5B, and 3E). This shift in the ratio of Class I to Class II COs is accompanied by a corresponding increase in genetic interference measured in two paired chromosome intervals (Fig 4C). We propose a model in which SRS2 acts on a subset of recombination intermediates channeled toward MUS81-dependent Class II COs – potentially by removing RAD51 to facilitate MUS81 access and/or promoting MUS81-mediated resolution. In this scenario, the absence of SRS2 would redirect these intermediates into the ZMM pathway, increasing Class I COs (supported by the moderate increase in MLH1/HEI10 foci in meiosis; Fig 6) while reducing Class II COs without altering the total number of CO precursors (Figs 3D and 5). Interestingly, such increase in MLH1 foci in UvrD-type

PLOS Genetics

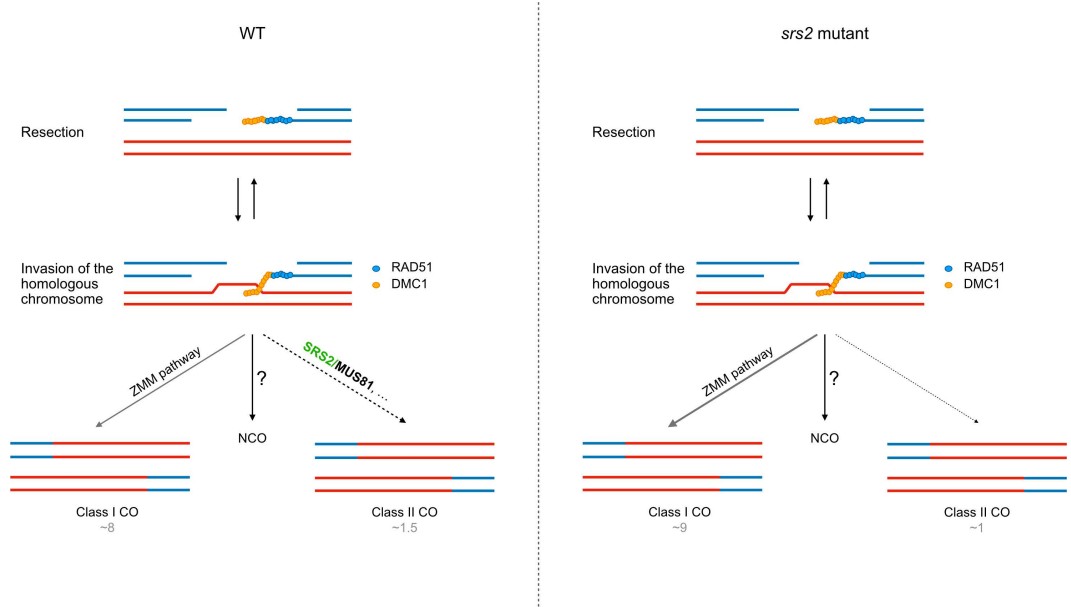

**Fig 6. Model for the role of SRS2 in meiotic recombination.** In contrast to yeast Srs2, Arabidopsis SRS2 does not appear to play a major role in RAD51 nucleofilament dynamics. Instead, SRS2 might stabilize a subset of recombination intermediates to facilitate their resolution by MUS81 (left panel). In *srs2* mutants (right panel), these intermediates would not be resolved by MUS81, and would instead be channeled into ZMM pathway, resulting in both an increase in Class I COs and a decrease in Class II COs.

mutants is not unique to Arabidopsis SRS2. In mouse, deletion of PARI, an antirecombinase related to yeast Srs2 and bacterial UvrD, also leads to increased MLH1 foci [60,61].

Previous studies in yeast have demonstrated that Srs2 interacts with and promotes Mus81-mediated resolution of recombination intermediates in somatic cells [57,62]. Interestingly, the stimulation of Mus81-Mms4 activity by Srs2 is independent of its helicase activity [62]. Moreover, Mus81 appears to prevent Srs2 from unwinding recombination intermediates, suggesting a tight coordination of the activities of both proteins for resolution of recombination intermediates. In meiosis, the non-interfering Class II CO pathway relies primarily on MUS81 in budding yeast, plants, and animals [4,5,49,51,63–65]. Accordingly, the absence of MUS81 leads to a 15–20% reduction in COs (Fig 5E). Similarly to our observation in *srs2* mutants, Arabidopsis *mus81* mutants exhibit reduced Class II COs and increased genetic interference [49,51]. If SRS2 facilitates MUS81-mediated resolution, the loss of SRS2 would impair MUS81 activity, leading to reduced Class II CO and increased genetic interference, as observed in Arabidopsis *srs2* mutants in this work (Fig 6). Interestingly, similar findings were reported for MUS81-deficient mice, where Class II COs suppression was associated with a significant increase in MLH1 foci, while the number of chiasmata per cell remained unchanged [65].

In conclusion, our data clearly point to a role for SRS2 in meiotic recombination, probably in stabilizing a subset of recombination intermediates for MUS81-mediated resolution into Class II COs. Further studies are nevertheless required to better understand the role of SRS2 in meiotic recombination and in particular in MUS81-dependent Class II COs establishment.

## Materials and methods

### Plant growth and *in vitro* culture conditions

In this study, the following mutant lines were used (all in Col0 background): *srs2–1* (GABI_637C01), *srs2–2* (GABI_647B04), *srs2–3* (SALK_039766), *zip4* (SALK_068052; [66]), *msh5* (SALK_110240; [67]), *mus81–2*

(SALK_107515; [49]), I1bc FTL (FTL567-YFP/ FTL1262-DsRed2/FTL992-AmCyan/*qrt1–2*) [68], I2fg FTL (FTL800-DsRed2/FTL3411-YFP/ FTL3263-AmCyan/*qrt1–2*) [68], *recq4a-4* (GABI_203C07; [69]), *rtel1–1* (SALK_ 113285; [70]), *rad54–1* (SALK_088057; [71]), RAD51-GFP [72].

Plants were grown on soil or *in vitro* (on 0.5X MS medium [M0255; Duchefa Biochemie] with 0.8% [m/v] agar, 1% sucrose) and stratified for 2 days at 4°C then grown in a growth chamber at 23°C under a 16h:8h light:dark photoperiod with 60% relative humidity. Seeds grown *in vitro* were first surface sterilized in 70% ethanol/0.05% SDS for 5 minutes, washed in 95% ethanol and then air-dried under a laminar flux hood.

### Protein structure prediction and comparison

We used AlphaFold Protein Structure Database [73,74] to predict the structure of AtSRS2 (AF-D1KF50-F1-v4) and ScSrs2 (AF-P12954-F1-v4). Both structures were visualized with ChimeraX [75].

### Molecular characterization of *srs2* T-DNA insertion mutants

The *srs2–1* mutant was genotyped using primers A and B to detect the wild-type loci and primers A and I or B and I (GABI-Kat T-DNA left border-specific primer) to detect the T-DNA insertion allele. For the *srs2–3* mutant, genotyping was performed using primers G and H to detect the wild-type loci and primers G (or H) and L (SALK T-DNA left border-specific primer) to detect the T-DNA allele. Left and/or right boundaries of T-DNAs have been sequenced by Sanger sequencing using the following primer pairs A+K and B+K for *srs2–1*, K+J for *srs2–2*, and G+L and L+H for *srs2–3* plants. Sequences of all primers used for genotyping and sequencing are listed in S3 Table.

### RT-PCR analyses

Total RNA was extracted on 7-days-old seedlings using RNeasy Plant mini kit (Qiagen). Two µg of RNA was then treated with RQ1 RNase-free DNase (Promega). Random hexamers were added to treated RNA and incubated 5 minutes at 70°C and then immediately placed on ice. Finally, M-MLV reverse transcriptase (Promega), 5X buffer, 10 mM dNTPs, 30U RNasin and ultrapure water were added and incubated at 37°C for 1h. PCR amplification were then performed using primers listed in S3 Table.

### MMC sensitivity assay

Surface sterilized seeds were sown onto half-strength MS medium (0.8% {m/v} agar, 1% sucrose) supplemented or not with 30 µM Mitomycin C (MMC; Sigma-Aldrich M0503). Dishes were stratified for 2 days at 4°C then grown at 23°C for 2 weeks. As described in [76], the number of true leaves per seedling was counted with a magnifying class to measure MMC sensitivity. Statistical analysis was performed with Kruskal-Wallis test (GraphPad Prism v10.4.0 software).

### Somatic homologous recombination assay using histochemical GUS staining

The frequency of somatic HR events was assessed by crossing the IU.GUS.8 line containing an interrupted *ß-Glucuronidase* (*GUS*) gene [77] with *srs2–1* and *srs2–3* mutants. Somatic HR events were analyzed in progeny homozygous for both the *srs2* mutation and the IU.GUS reporter.

Surface sterilized seeds were grown *in vitro* on MS medium (2 days of stratification then 2 weeks at 23°C), then incubated in GUS staining buffer (0.2% Triton X-100, 50 mM sodium phosphate buffer, pH 7.2, and 2 mM X-Gluc [Biosynth] dissolved in N,N-dimethylformamide). Plants were then vacuum-infiltrated for 15 minutes and then incubated at 37°C for 24h. Staining buffer was then replaced with 70% EtOH to remove leaves pigmentation. Finally, blue spots were counted under a binocular microscope. Statistical analysis was performed with Kruskal-Wallis test (GraphPad Prism v10.4.0 software).

### RAD51 foci on root apex nuclei

Plants were grown for 5 days on half-strength MS medium and fixed in 4% paraformaldehyde (in 1X PME buffer (50 mM PIPES (pH 6.9), 5 mM MgSO4, 1 mM EGTA)) for 45 min. Immunostaining in root tip nuclei was then performed as previously described [78]. Slides were incubated with rat α-RAD51 (1/500 in 3% BSA, 0.05% Tween-20 in 1X PBS [79]) in a moist chamber at 4°C overnight. Slides were washed 3 times in 1X PBS-0.05% Tween-20, air-dried, then incubated with secondary antibody solution (chicken anti-rat Alexa 488 (Invitrogen) diluted 1/1000 in 3% BSA, 0.05% Tween-20 in 1X PBS) in a moist chamber for 3h at room temperature in the dark. Slides were finally washed 3 times in 1X PBS-0.05% Tween-20, air-dried, and mounted in VECTASHIELD mounting medium containing DAPI (1.5 μg/ml; Vector Laboratories Inc.). Z-stacks images were acquired with a Zeiss Cell Observer Spinning Disk microscope and analyzed using Imaris software v9.8.2 as previously described [80]. Briefly, 3D root nuclei were segmented and a mask was created on the segmented surfaces to display EGFP (Alexa488, marking RAD51 foci) only in the surfaces. A random color mask was applied on the DAPI channel to assign a unique color ID to each surface. Finally, spots were created with the "Spots" tool on RAD51 foci using Sum Square of Alexa488 as a quality control. Spot size was standardized at 0.3 μm in diameter, and 0.5 μm z-axis elongation. Statistics (Surface: Volume, Median Intensity; Spots: Intensity Min, Max, Mean, Median, Sum, SD, Sum Square, Median Intensity) were exported and data were plotted using GraphPad Prism v10.4.0. Statistical analysis of the number of RAD51 foci per nucleus was performed using Kruskal Wallis test (GraphPad Prism v10.4.0 software).

### Plant fertility assessment

Plant fertility was assessed by counting the number of seeds per silique. About 30 siliques from the primary stem were collected per plant and bleached in 95% EtOH at 70°C for several hours. The number of seeds per silique was counted manually under a binocular microscope. All analyzed plants were grown in the same conditions. Statistical analysis was performed using Kruskal-Wallis test (GraphPad Prism software v10.4.0).

### Pollen viability test

Pollen viability was measured using Alexander staining [81]. Opened flowers were collected and dissected on a microscope slide in 10 μl of Alexander solution (containing 1% malachite green, 1% acid fuchsin, and 1% orange G). Viable pollen grains stained in purple, while non-viable (dead) pollen grains stained green. For each genotype, an average of 2000–4000 pollen grains were examined.

### Meiotic chromosome spreads

Meiotic chromosome spreads were prepared as previously described [82]. Inflorescences were collected from secondary stems and fixed 3 times 30 min in Carnoy's fixative (3:1 EtOH: acetic acid). Fixed inflorescences were washed once in ultrapure water then twice in 10 mM citrate buffer (pH 4.5). Flower buds were then digested in enzyme mixture (0.3% cellulase, 0.3% pectolyase, and 0.3% cyclohelicase; Sigma-Aldrich) for 3h at 37°C in a moist chamber. The digestion reaction was stopped by placing the slides on ice and replacing enzyme mix with ice-cold 10 mM citrate buffer (pH 4.5). Immature flower buds of appropriate stage (0-3-0.6 mm) were selected under a binocular microscope, placed individually on a clean microscope slide, and crushed with a dissection needle. Chromosomes were spread by stirring for 1min in 20 μl of 60% acetic acid at 45°C, fixed with Carnoy's fixative, and air-dried. Finally, slides were mounted in VECTASHIELD mounting medium containing DAPI (1.5 μg/ml DAPI; Vector Laboratories Inc.) and covered with 24 × 32 mm coverslip. Images were acquired with a Zeiss AxioImager.Z1 epifluorescence microscope equipped with an Axio-Cam Mrm camera and DAPI filter.

### Immunolocalization of MLH1/HEI10 on meiotic chromosome spreads

All slides used in this experiment were washed in acetone, water, and EtOH then airdried before use. Immunolocalization of MLH1 and HEI10 proteins was performed on meiotic chromosome spreads slides containing diakinesis-staged

meiocytes as previously described [83]. Selected slides were placed in room-temperature PBST (1X PBS – 0,1% Triton), transferred for 45 seconds in (barely) boiling Tris-sodium citrate solution, then transferred back in room-temperature PBST. Slides were then incubated with the primary antibody solution (Rabbit MLH1: 1/200 [83]; Chicken HEI10: 1/100 [84]; diluted in 1% BSA-PBST) for 2 days in a moist chamber at 4°C. Slides were then washed 3 times in PBST and incubated with the secondary antibody solution (Donkey anti-Rabbit Cy3 (Jackson ImmunoResearch, ref. 711-165-152): 1/1000; goat anti-chicken Alexa488 (Invitrogen, Life Technologies; ref. A11039): 1/1000; diluted in 1% BSA-PBST) for 30 min in a dark moist chamber at 37°C. Finally, the slides were washed 3 times in PBST and air-dried in the dark for 5 minutes. Slides were mounted in Vectashield mounting medium containing DAPI (1.5 µg/ml DAPI; Vector Laboratories Inc.) and covered with a coverslip. The coverslip was sealed to the slide using clear nail polish.

### RAD51/ASY1 and RAD51/ZYP1 immunolocalization on meiocytes

All slides used in this experiment were washed in acetone, water, and EtOH then air-dried before use. Inflorescences were harvested from secondary stems and placed in an ice-cold Petri dish with moist filter paper. Buds ranging from 0.3 mm to 0.5 mm were selected using a binocular microscope. Approximately 6–8 buds were dissected in 10 µl Enzyme Mix (0.4% Cytohelicase, 1.5% sucrose, 1% PVP, in H2O) to retrieve as many anthers as possible. After a 5 minutes digestion at 37°C in a moist chamber, anthers were macerated with a dissection needle, 10 µl Enzyme Mix was added and slides were again incubated at 37°C for 10 minutes in a moist chamber. Then, 10 µl of 1% Lipsol was added to the slide and the droplet was stirred vigorously for 2 minutes, then incubated for 3 minutes. Under a fume hood, 40 µl of 4% PFA was added and spread with a needle. Slides were air-dried for 2–3 hours.

For the immunostaining, slides were washed 3 times in PBST (1X PBS, 0.1% Triton X-100), and blocked with 1% BSA (in PBST) at room temperature for 30 minutes. The slides were then incubated with the primary antibody solution (rat anti-RAD51: 1/500 [79]; Guinea Pig anti-ASY1: 1/500 [85]; Rabbit anti-ZYP1: 1/500 [86]; diluted in 1% BSA in PBST) for 2 days in a moist chamber at 4°C. Then, the slides were washed 3 times in PBST and incubated with the secondary antibody solution (donkey anti-rat Cy3 (Jackson ImmunoResearch, ref. 712-165-150): 1/1000; goat anti-G. Pig Alexa488 (Invitrogen, ref. A-11073): 1/1000; diluted in 1% BSA in PBST) for 1 hour at 37°C in a dark moist chamber. Finally, slides were washed 3 times in PBST and air-dried, and mounted in Vectashield mounting medium containing DAPI (1.5 µg/ml DAPI; Vector Laboratories Inc.) and covered with a coverslip. The coverslip was sealed to the slide using clear nail polish.

### Recombination measurement and interference using FTLs

We used Fluorescent Tagged Lines (FTLs) to estimate male meiotic recombination rates at two genetic intervals: I1bc on chromosome 1 and I2fg on chromosome 2. Heterozygous plants for the linked fluorescent markers were generated and siblings from the same segregating progeny were used to compare the recombination frequency between different genotypes. Slides and fluorescent tetrad analysis were performed as previously described [48]. Tetrads were counted and attributed to specific classes (A to L) (see Figs 4 and S5 the classification and raw data). Genetic distances of each interval were calculated using Perkins equation as follows: $X = 100 \ [(1/2 \ \text{Tetratype} + 3 \ \text{Non-Parental Ditype})/n]$ in cM [48]. Interference was estimated using the Coefficient of Coincidence $\left( \text{CoC} \ (I1, I2) = \frac{\text{frequency of double CO in I1 and I2}}{\text{frequency of CO in I1} * \text{freq of CO in I2}} \right)$, and interference calculated as 1-CoC. Thus, a calculated interference of 0 reveals the absence of interference in the interval. Statistical analysis was performed using Z test, and data plotted using GraphPad Prism software v.10.4.0.

### Metaphase I image analysis

Chiasma count of metaphase I-stage meiocytes was performed as previously described in [87]. Statistical analysis of chiasmata count of metaphase I-staged meiocytes was performed using an unpaired t-test (GraphPad Prism software v. 10.4.0). Meiotic chromosome spreads pictures obtained for the counting of bivalents in ZMM mutant background were assigned a random name to allow an unbiased analysis using the Blind Analysis Tool (Fiji). Statistical analysis of the number of bivalents was then performed using unpaired Mann-Whitney test (GraphPad Prism software v.10.4.0).

## Statistical analyses

The normality of data distributions was assessed using D'Agostino–Pearson test. For comparisons between two independent datasets, an unpaired Student's t-test was used if both followed a normal distribution, otherwise a Mann–Whitney test was applied. For comparisons involving more than two datasets, the Kruskal–Wallis test followed by Dunn's post hoc test for multiple comparisons was performed. When comparing datasets that included multiple experimental conditions, a two-way ANOVA was used. FTL data was analyzed using Z-test.

## Supporting information

**S1 Fig. Sequence alignment of ScSrs2 and AtSRS2 and alignment of UvrD, ScSrs2, AtSRS2, and PpSRS2 conserved helicase motifs.** (A) Identical (red), conserved (green) and semi-conserved (yellow) amino acids are depicted. Magenta and turquoise arrows show UvrD-like helicase ATP-binding and UvrD-like helicase C-terminal domains, respectively. Identity 26,26% (219/ 834), similarity 43,53% (363/ 834). (B) Alignment of the helicases domain protein sequences of UvrD, yeast Srs2, *A. thaliana* SRS2, and *Physcomitrella patens* SRS2. Comparison of the amino acid residues of the different conserved helicase domains shown to be essential for ScSrs2 function are underlined in bold. Three of these amino acids are not conserved (neither identical nor similar) in Arabidopsis and *P. patens* and are marked with an asterisk. The amino acids indicated by arrows correspond to those of Arabidopsis. Y775 in yeast corresponds to V963 in Arabidopsis and V1120 in *P. patens.*
(PDF)

**S2 Fig. Arabidopsis SRS2 T-DNA Insertion Mutants.** (A) Structure of SRS2 and T-DNA insertion mutant alleles. Dark grey boxes show exons and light grey boxes indicate 5' and 3' untranslated regions. The position of the T-DNA insertions is indicated (purple, blue, or green triangles) with arrows showing the orientation of the left border and sequences of the T-DNA/chromosome junctions below (SRS2 sequence in black and T-DNA sequence in red). In *srs2–1*, insertion is accompanied by a 32 bp deletion in exon 7. A putative in-frame TGA codon is underlined. Numbering under the sequences is relative to the SRS2 start codon. (B) Sequences of the T-DNA/ chromosome junctions in *srs2–2* and *srs2–3*. Putative in-frame stop codon is underlined. Numbering under the sequences is relative to the SRS2 start codon. (C) RT-PCR analyses of transcripts of srs2 insertion mutants. Amplification of the actin transcript (Actin RT+) was used as a control for RT-PCR. The positions and orientations of the PCR primers are shown with capital letters in the diagram.
(PDF)

**S3 Fig. RAD51 foci quantification in root apex nuclei.** Quantification of RAD51 foci in root tip nuclei of WT, *srs2–1,* and *srs2–3* mutant lines before and after MMC treatment. Data are shown as mean ± SD, with n indicating the number of cells analyzed. More than 300 nuclei from at least 3 seedlings were analyzed per genotype. Cells analyzed are the same as in Fig 2G. P-values were calculated using nonparametric statistical analysis (Kruskal–Wallis test); * p-value < 0.05; **** p-value < 0.0001.
(PDF)

**S4 Fig. Meiotic progression in *dmc1 srs2* double mutant.** Meiotic progression of male meiocytes stained with DAPI in WT, *dmc1*, and *dmc1 srs2–1* plants. Absence of DMC1 leads to asynapsis (late Prophase I) and lack of inter-homologue CO. Intact univalents are thus observed at Metaphase I owing to DSB repair by RAD51, most probably using sister chromatids. Univalents then segregate randomly at Anaphase I and ultimately this produces unbalanced Tetrads. A similar meiotic phenotype is observed in *dmc1 srs2* mutant. Scale bar: 10 μm.
(PDF)

**S5 Fig. CO frequency and interference in *srs2–3* mutant.** (A) CO frequency of WT and *srs2–3* mutant within I1bc interval represented as the genetic distance (cM). Each dot represents one plant, with 400–800 tetrads analyzed per

plant. Mean is presented as a red bar. (B) Interference within I1bc interval of WT and *srs2–3* plants. Each dot represents one plant. Mean is presented as a red bar. Statistical analysis was performed using Z-test.
(PDF)

**S6 Fig. Increased MLH1 foci in *srs2–1* and *srs2–3* mutants.** (A) Representative images of MLH1 immunostaining on diakinesis-staged male meiocytes in WT and *srs2–1* mutant. Scale bar: 10 µm. (B) Number of MLH1 foci per cell in *srs2–1* and *srs2–3*, with n indicating the number of cells analyzed. Data are represented as mean (red line), with each dot representing one individual cell. Statistical analysis was performed using Mann-Whitney test. * p-value < 0.05.
(PDF)

**S1 Table. FTLs raw data and Interference ratio calculation for I1bc and I2fg intervals.**
(XLSX)

**S2 Table. Raw data for foci quantification of MLH1 immunolocalization.**
(XLSX)

**S3 Table. List of primers used for characterizing *Atsrs2* T-DNA mutants.**
(DOCX)

**S1 Data. Raw data for all countings presented in this study.**
(XLSX)

## Acknowledgments

We thank Mathilde Grelon for ZYP1, MLH1 and HEI10 antibodies, Peter Schlögelhofer for RAD51 antibody and Eugenio Sanchez-Moran for the ASY1 antibody. We also acknowledge Christophe Lambing for providing the *zip4* and *msh5* mutant seeds, the CLIC platform (Clermont-Ferrand Imagerie Confocale, Clermont Auvergne University) for help in spinning-disk microscopy, and all members of the recombination group for their help and discussions.

## Author contributions

**Conceptualization:** Charles I WHITE, Olivier Da Ines.

**Formal analysis:** Valentine PETIOT, Floriane CHÉRON, Olivier Da Ines.

**Funding acquisition:** Charles I WHITE, Olivier Da Ines.

**Investigation:** Valentine PETIOT, Floriane CHÉRON, Olivier Da Ines.

**Methodology:** Valentine PETIOT, Olivier Da Ines.

**Project administration:** Charles I WHITE, Olivier Da Ines.

**Supervision:** Olivier Da Ines.

**Writing – original draft:** Valentine PETIOT, Olivier Da Ines.

**Writing – review & editing:** Valentine PETIOT, Charles I WHITE, Olivier Da Ines.

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
