## [Decision Letter · Decision Letter 0]

9 Apr 2025

PGENETICS-D-25-00237

Dual role of Arabidopsis SRS2 helicase in meiotic recombination

PLOS Genetics

Dear Dr. Da Ines,

Thank you for submitting your manuscript to PLOS Genetics. After careful consideration, we feel that it has merit once appropriate modifications and possibly addtions of new data are made. Therefore, we invite you to submit a revised version of the manuscript that addresses the points raised during the review process.

Please submit your revised manuscript within 60 days Jun 08 2025 11:59PM. If you will need more time than this to complete your revisions, please reply to this message or contact the journal office at plosgenetics@plos.org. Please include the following items when submitting your revised manuscript:

We look forward to receiving your revised manuscript.

Kind regards,

Tomo Kawashima

Academic Editor

PLOS Genetics

Anne Goriely

Editor-in-Chief

PLOS Genetics

Aimée Dudley

Editor-in-Chief

PLOS Genetics

Anne Goriely

Editor-in-Chief

PLOS Genetics

**Additional Editor Comments (if provided):**

All three reviewers are enthusiastic about your work. To strengthen your argument regarding the function of SRS2, we kindly ask you to carefully read the reviewers' comments and address them by explaining in a more concise manner, providing additional information to further validate your claims, or rephrasing or toning down some statements where appropriate. If you need any clarification or assistance, don’t hesitate to contact me. I’m here to help ensure the review process goes smoothly.

**Journal Requirements:**

**Reviewers' comments:**

Reviewer's Responses to Questions

**Comments to the Authors:**

Reviewer #1: The authors present their findings on the role of the SRS2 helicase in both somatic and meiotic cells in Arabidopsis thaliana. Interestingly, they show a different role of SRS2 in somatic cells than what has been showed in yeast. The authors also study the role or SRS2 in meiotic cells, where they show its involvement in both class I and class II crossover regulation. Particularly, their data shows that SRS2 is needed for the formation of class II crossovers, when, to date, only few actors have been shown to be involved in this pathway.

Major comments:

1. Changes in the strength of interference are not enough to claim that interference (between class I COs) is changed. The addition of class II crossover in the mix of class I can lead to apparent changes in interference strength without affecting the interference mechanism per se. For instance, Berchowitz et al 2007 show that in mus81 mutant interference increases. Conversely, in mutants with elevated class II crossover count, interference is decreased (Crismani et al., 2012, Girard et al., 2014 etc…). It is very likely that the genetic interference measured in srs2 could be a consequence of increasing the class II pathway, and not that SRS2 is a positive effector of class I COs. In the Discussion, this is actually how the authors phrase it (lines 388 and 411), but in the results, the wording can lead to confusion (lines 255-274). If the authors wish to claim that srs2 is responsible for an increase in genetic interference (between class I COs), then they should measure the CoC in a srs2 mus81 double mutant. The expectation is that if there is a direct effect of SRS2 in interference, the CoC would be higher in the double mutant compared to the single mus81 mutant. Otherwise, given the epistasis between SRS2 and MUS81 the increase in genetic interference should be discussed as being most likely due to a decrease in class II crossovers. This is actually a good argument for the decrease of class II COs that the author could capitalize on (in addition to fig 6 results). I would therefore strongly recommend showing the interference measure after the data with mus81 (fig 6) to avoid any confusion.

2. Figure 7: The defects shown in the paper are very moderate. Therefore, in the model in Fig. 7, the ++ and - - effects in the srs2 mutant could be toned down (to + and -, indicating a slight increase and a slight decrease). Same for the thickness of the arrows, the data show an increase of 1 class I CO, which does not deserve such a strong arrow.

- Figure 7: Additionally, considering their model, it feels like the authors are claiming that SRS2 would act directly on the loading/dynamic of RAD51/DMC1. Based on data from yeast, this is plausible. However, the presented data does not support this hypothesis more than an alternative one where the turnover of RAD51 foci is altered due to some defects in repair. If the authors want to indeed claim a direct role of SRS2 on RAD51 filament, they should at least quantify the RAD51 foci during meiosis and not only in somatic cells, where only a slight increase was shown. Otherwise, they should remain vague in the figure.

Minor comments/manuscript organization:

1. Generally, it would be nice to have conclusions as Figure titles (2 / 3 / 4), or at least making the titles a little bit more informative

2. Figure 1 could be in supplementary figures as it is not a major result but simply a verification of T-DNA insertions. Perhaps it would be more interesting to show the (predicted) structures of yeast Srs2 and plant SRS2 proteins, with the conserved and non-conserved residues, with a particular focus on their putative functionality (as discussed in the Discussion section)

3. In Fig2 and Fig3 the authors show that the srs2 mutants are not more susceptible to DNA damage, which, interestingly, is different from what has been shown in yeast. Is SRS2 expressed in somatic cells in Arabidopsis? Are there paralogs of srs2 in the Arabidopsis genome? Could there be some redundancy?

4. In Fig3C, the data layout makes it difficult to see any differences between the genotypes, and Fig3D is redundant (same type of data). Perhaps it could be enough to show that RAD51 focus number increases upon treatment with MMC in srs2 mutants using only Fig3D (and keep 3C for supp data).

5. I would recommend fusing Fig2 and 3 (it would become figure 1) to showcase the differences in requirement for SRS2 in mediating somatic repair in response to DNA damaging agents. This could be accompanied, if the authors chose to, by the fusion of section 2 and 3 of the text. The main message is that Srs2 is dispensable for DNA repair and RAD51-dependent recombination in somatic cells, unlike in yeast, which is interesting.

6. Fig5 srs2-3-containing data could be relegated to a supplementary figure. However, FigS3 should absolutely be together with l1bc interval data for srs2-2 mutant. This would better reflect the reality of the data (slight change in interference strength)

7. Fig5: Why showing C/D + E/F? Is the difference between the two wild-type controls in the two experiments (11 foci in D vs 8 co-foci in E) something we should take into account? Perhaps merging MLH1 data and ignore the Hei10, or just show the number of co-localised MLH1/HEI10 foci?

Typos:

- Line 80: “Srs2 (Suppressor of RAD Six-screen mutant 2)” and remove (Suppressor of RAD Six-screen mutant 2) from line 85

- Line 94: Remove parenthesis in “SDSA” and add “Synthesis-Dependent Strand Annealing”

- Line 106: remove “single-stranded DNA” since ssDNA has been defined before

- Line 127: in plants

- Line 131: In vitro

- Line 155: a homolog of the yeast Srs2 was found

- Line 159: Remove “(SRS2)”

- Line 188: Remove “strand”

- Line 222-224: The deviation does not match the deviation written on Fig3C

- Line 244-246: Split in two sentences

- Line 294: Replace “In accordance with expectation,” by “As expected,”

- In Materials and Methods some antibody references are missing (e.g., a-RAD51 in line 483; Rabbit MLH1 and Chicken HEI10 in line 534, donkey a-rabbit Cy3 and goat a-Chicken Alexa 488 in lines 536-537; guinea pig a-ASY1, rabbit a-ZYP1 in line 558; donkey a-rat Cy3 and goat a-guinea pig alexa 488 in lines 560-561)

- Fig1A legend: In line 848-849 remove “Triangles (purple, blue, or green triangles).” (not a sentence)

- Fig1C legend: In line 857 remove ACT and use just Actin RT+ as shown in the figure.

- Figure2B legend: Missing total number of seedlings analyzed. Meaning of p-value **** and * missing

- Figure2C legend: Missing total number of seedlings analyzed.

- Figure3D legend: Missing meaning of “ns” (non-significant)

- Fig2C missing which T-DNA insertion srs2 mutant was used in the double mutants

- Fig5B: Scale should range from 0 to 0.8 and not 0.4 to 0.8

- Throughout the figures, please homogenize the statistics to plot the mean + SD and not the mean + SEM in all plots.

All the best to all authors for the revision !

Reviewer #2: The manuscript by Petiot et al. describes their investigations into the in vivo role of the SRS2 homolog from Arabidopsis thaliana, a helicase that plays an important role in homologous recombination by regulating the strand invasion process in yeast. In plants (Arabidopsis), a previous in vitro/biochemical study demonstrated that SRS2 has a role in homologous recombination processes, but its function in plants remains unknown. In the current manuscript, these authors utilized multiple null alleles of SRS2 to characterize this gene in Arabidopsis. They conducted detailed experiments to demonstrate that, in contrast to yeast studies, AtSRS2 is dispensable for somatic homologous recombination, meiosis progression, and fertility in Arabidopsis. However, through an impressive set of cytological observations, quantifications, and recombination assays using fluorescent-tagged lines, the authors show that AtSRS2 plays a pro- and anti-crossover role in meiosis. The following are my comments on the manuscript.

The authors use the fluorescent-tagged lines/markers (FTLs) in a tetrad mutant background to score for recombinants. This well-established assay is, in general, very informative. However, the authors should provide more information on the marker positions on the chromosomes and in the context of centromeres (e.g., provide a schematic with physical distances). Centromeres are known to negatively affect recombination, and this is not discussed concerning markers on studied chromosome 1 or 2. Are those loci studied on the chromosome potentially affected by centromeres?

One of the major claims of the authors regarding the role of SRS2 in recombination is based on the results from chromosome 1. The findings on chromosome 2 are not very convincing in the context of interference, given that the authors state in line 271, “.... not statistically significant, likely due to the very short size of the I2fg interval, which in practice limits the detection of a sufficiently large number of double CO events. If the authors believe that the short size of the I2fg interval limits the occurrence of significant double CO events, then they might consider the FTLs on other chromosomes or possibly collect data by sequencing multiple sets of tetrad progenies.

Recombination and crossover frequencies differ between male and female meiosis. Do the authors think the SRS2 might have a differential role in these processes?

Given the authors' expertise in immunolocalizing crucial proteins involved in Class I and Class II crossovers, the localization of AtSRS2 in meiosis and mitosis is explicitly missing. Can the authors demonstrate that it is indeed physically part of the recombination process either by colocalization or showing its interaction with one or more Arabidopsis recombination machinery components in vivo?

Regarding SRS2's pro-CO role, is there a way to demonstrate that the lack of SRS2 results in rechanneling the NCO to the ZMM pathway? E.g., overexpression of AtSRS2 in meiosis?

Although the authors raise concerns about the lack of conservation of ScSrs2 Y775 (V963 in Arabidopsis) related to “potential functional divergence or loss ” in both the results and discussion sections of this manuscript, its relevance to the proposed role (pro and anti CO?) remains unclear from the discussion. Additionally, the authors cite a previous study on SRS2 in Physcomitrella. A comparison of AtSRS2 vs. PpSRS2 in the context of ScSRS2 amino acid conservation (Y775) and other sites could be more insightful.

Incorporating the tetrad analysis data into the main figure (with all types of crossovers, especially double crossovers relevant to interference, highlighted) will allow readers to better appreciate the subtle yet intriguing role of AtSRS2 in meiosis.

Figure 3: Rad51 foci per nuclei. Do you have any thoughts on why the majority of cells do not exhibit RAD51 foci after MMC treatment? However, when they are observed, most RAD51 foci-positive nuclei show 10 or more signals. Is a specific cell cycle stage more vulnerable to MMC treatment than others? What threshold size was used to call a given fluorescent signal as a RAD51 signal with respect to any potential background signals?

Figures 3A, 3C, 3D, and lines 218-219: I see a few RAD51 foci in the charts, so please change the statement accordingly. My understanding is that there is a background level of somatic HR that is RAD551 but may not be efficiently captured.

Figure 4: Fertility in Arabidopsis can be measured in various ways, but it is often not done or reported accurately. The number of seeds per silique or the observation of meiotic abnormalities through cytology seem to be reasonable measures, but they lack stringency when the events are relatively rare yet biologically relevant (not spontaneous) compared to WT. One additional estimate on the female side could be the count of unfertilized ovules in matured or developing siliques (represented by tiny, dried, whitish ovules) and aborting ovules after fertilization (brownish ovules in matured or developing siliques). On the male side, pollen viability can be assessed through staining, such as using Alexander staining on mature, intact anthers. If the abnormalites are relatively scarce in the srs2 mutant background, counting the chiasmata in 45 or 66 meiocyte spreads, as reported in this manuscript, may not serve as an ideal assay.

Lines 159-160: What do the authors mean by all the amino acids recently identified as essential? This is not entirely clear.

Figure 2 B: Any thoughts of a significant increase in sensitivity to MMC 40uM concentration by WT strains compared to mutants?

Figure 2 C: Which srs2 mutant alleles were used in single and double mutant studies? Only one allele is marked (srs2-1), while the others are not identified. Why do the number of true leaves in the control (WT) and other mutants/double mutants exceed those in Figure 2B? Why is there this discrepancy?

Figure 2D; Include some representative GUS staining images and the interrupted GUS construct’s schematic to aid readers.

Expand SDSA at its first mention in the manuscript

Lines 278-280: sentences appear convoluted; paraphrasing would be more helpful to readers

Figure 1 Legend: fix the repetition of triangles

To aid readers, label the amino acids mentioned in lines 161-166 with specific coordinates on Figure S1.

Reviewer #3: Petiot et al seek to understand the in vivo role of SRS2 helicase in Arabidopsis meiotic recombination. AtSRS2 has been shown to possess helicase activity in vitro in prior studies. The authors in this work find a dispensable role for AtSRS2 in mitotitc recombination while a shift in the balance between class I COs and class II COs is observed in the absence of SRS2. The data presented shows a role for SRS2 similar to that of MUS81. The manuscript overall is well written with the data presented as elegant figures.

Major questions/comments

Why the authors performed the non-parametric Kruskal-Wallis test is not clear? Are the data analyzed not normally distributed? In an exception to this, two other tests were performed in Fig. 4 and Fig. 6. Is there an explanation to this? This will affect a lot of the data interpretation in this study.

An important question is about the proposed “dual role” for SRS2 in Arabidopsis. The idea of a “dual role” for SRS2 is not very convincing. The data presented shows that Atsrs2 mutant has reduced class II CO (Fig) and an increased class I CO and genetic interference (Fig). This data is similar to the observation for mus81 mutant (lines 408-409). Further, the authors show that mus81 is epistatic to srs2 and indeed the triple mutant zip4srs2mus81 is not different from zip4srs2 or zip4mus81. All of these point to the fact that SRS2 functions similarly and probably together with MUS81 in promoting/balancing class II CO. If this is the case, would the authors also argue that MUS81 has a “dual role”?

The idea of shifting the balance between Class I COs and Class II Cos for both SRS2 and MUS81 is more convincing.

The model presented in Fig .7 could be improved by adding more details. Currently, the visuals don’t convey a lot. For example,” what is the “++” supposed to mean in the right panel next to the cartoon of RAD51 and DMC1? Adding details to the graphic will convey the message better. Additionally, can the authors include an estimate of the number of class I CO and class II CO in the right panel from the data obtained in this work?

Minor suggestions:

Cite figures in “discussion”

Specific details:

Line 94: explain SDSA at first mention

Are there other UvrD-type DNA helicases in Arabidopsis, such as an FBH1 like (lines 115-121)?

Line 462: expand MMC and provide catalog number, if possible

Line 468: how were the lines generated? Briefly, include the crossing scheme in a few words.

Lines 481-482: expand PFA. What is the molar concentration corresponding to “1x”?

Line 483: what is the source of anti-RAD51?

Line 513: for how long?

Lines 848-849: “Triangles…” is redundant

Line 869: data are from how many experiments is missing for panel C. This information is provided for panel B.

Line 873: similar comment as above.

Line 957: Giving species abbreviation before the protein would be helpful, such as ScSrs2, etc.

Lines 962-963: what do the authors mean by “Srs2 activities”? It is counterintuitive that the essential amino acids are not conserved but the Arabidopsis SRS2 has in vitro helicase activity. Could it be written in a different way?

**Have all data underlying the figures and results presented in the manuscript been provided?**

Reviewer #1: Yes

Reviewer #2: Yes

Reviewer #3: Yes

PLOS authors have the option to publish the peer review history of their article (what does this mean? ). If published, this will include your full peer review and any attached files.

**Do you want your identity to be public for this peer review?** For information about this choice, including consent withdrawal, please see our Privacy Policy .

Reviewer #1: No

Reviewer #2: No

Reviewer #3: No

**Figure resubmission:**
---

## [Decision Letter · Decision Letter 1]

24 Jul 2025

Dear Dr Da Ines,

We are pleased to inform you that your manuscript entitled "Role of Arabidopsis SRS2 helicase in MUS81-dependent Class II CO formation" has been editorially accepted for publication in PLOS Genetics. Congratulations!

Yours sincerely,

Tomo Kawashima

Academic Editor

PLOS Genetics

Anne Goriely

Editor-in-Chief

PLOS Genetics

Aimée Dudley

Editor-in-Chief

PLOS Genetics

Anne Goriely

Editor-in-Chief

PLOS Genetics

Comments from the reviewers (if applicable):

Reviewer's Responses to Questions

**Comments to the Authors:**

Reviewer #1: The authors took into account all of our previous comments, and we recommend this manuscript for publication.

Couple very minor comments (that do *not* need to be reviewed again):

- Title "The absence of SRS2 affects MUS81-dependent Class II COs" : affects does not mean much. We propose "SRS2 is required for MUS81-dependent CO formation in zmm mutants"

- line 240 : authors refer to Figure 3C but they mean Figure Supp 3C

Reviewer #2: The authors made a marked improvement to the manuscript compared to the initial submission and attempted to demonstrate that AtSRS2 plays a role in Class II crossover regulation. I have the following minor comments on this manuscript.

The AlphaFold model does not add any significant value, as shown in Figure 1 of this manuscript. Consider moving it to the supp. Figures.

In the abstract, end of introduction, and in discussion, instead of the phrase "some recombination intermediates," a subset of recombination intermediates may be used

In the discussion line 432, "potentially by removing RAD51 to facilitate MUS81" appears to contradict their earlier conclusion (Lines 282-284), "This suggests that SRS2 has no major effect on RAD51-mediated meiotic DSB repair in Arabidopsis meiosis."

Discussing Line 455: The cited reference "64" pertains to yeast. Are the authors intended to reference Holloway et al. (67)?

Reviewer #3: The authors have satisfactorily revised the manuscript, and it is suitable for publication in the current form

**Have all data underlying the figures and results presented in the manuscript been provided?**

Reviewer #1: Yes

Reviewer #2: Yes

Reviewer #3: Yes

PLOS authors have the option to publish the peer review history of their article (what does this mean? ). If published, this will include your full peer review and any attached files.

**Do you want your identity to be public for this peer review?** For information about this choice, including consent withdrawal, please see our Privacy Policy .

Reviewer #1: **Yes: ** Chloé Girard

Reviewer #2: No

Reviewer #3: **Yes: ** Siddique I. Aboobucker

**Data Deposition**

http://datadryad.org/submit?journalID=pgenetics&manu=PGENETICS-D-25-00237R1

**Press Queries**

---

## [Editor Report · Acceptance letter]

PGENETICS-D-25-00237R1

SRS2 is required for MUS81-dependent CO formation in *zmm* mutants

Dear Dr Da Ines,

We are pleased to inform you that your manuscript entitled " 

SRS2 is required for MUS81-dependent CO formation in *zmm* mutants

" has been formally accepted for publication in PLOS Genetics! Your manuscript is now with our production department and you will be notified of the publication date in due course.

With kind regards,

Lilla Horvath

PLOS Genetics

On behalf of:
